# Intensity Thresholds for External Workload Demands in Basketball: Is Individualization Based on Playing Positions Necessary?

**DOI:** 10.3390/s24041146

**Published:** 2024-02-09

**Authors:** Sergio J. Ibáñez, Carlos D. Gómez-Carmona, Pablo López-Sierra, Sebastián Feu

**Affiliations:** 1Research Group in Optimization of Training and Sports Performance (GOERD), Department of Didactics of Music Plastic and Body Expression, Faculty of Sport Science, University of Extremadura, 10003 Caceres, Extremadura, Spain; sibanez@unex.es (S.J.I.); plopezp@alumnos.unex.es (P.L.-S.); sfeu@unex.es (S.F.); 2BioVetMed & SportSci Research Group, International Excellence Campus “Mare Nostrum”, Faculty of Sport Sciences, University of Murcia, 30720 San Javier, Murcia, Spain

**Keywords:** team sports, kinematics, impacts, player load, inertial devices

## Abstract

Currently, basketball teams use inertial devices for monitoring external and internal workload demands during training and competitions. However, the intensity thresholds preset by device manufacturers are generic and not adapted for specific sports (e.g., basketball) and players’ positions (e.g., guards, forwards, and centers). Using universal intensity thresholds may lead to failure in accurately capturing the true external load faced by players in different positions. Therefore, the present study aimed to identify external load demands based on playing positions and establish different intensity thresholds based on match demands in order to have specific reference values for teams belonging to the highest competitive level of Spanish basketball. Professional male players (*n* = 68) from the Spanish ACB league were monitored during preseason official games. Three specific positions were used to group the players: guards, forwards, and centers. Speed, accelerations, decelerations, impacts/min, and player load/min were collected via inertial devices. Two-step clustering and k-means clustering categorized load metrics into intensity zones for guards, forwards, and centers. Guards covered more distance at high speeds (12.72–17.50 km/h) than forwards and centers (*p* < 0.001). Centers experienced the most impacts/min (*p* < 0.001). Guards exhibited greater accelerations/decelerations, albeit mostly low magnitude (*p* < 0.001). K-means clustering allowed the setting of five zones revealing additional thresholds. All positions showed differences in threshold values (*p* < 0.001). The findings provide insights into potential disparities in the external load during competition and help establish position-specific intensity thresholds for optimal monitoring in basketball. These data are highly applicable to the design of training tasks at the highest competitive level.

## 1. Introduction

Basketball is an intermittent high-intensity sport characterized by repeated explosive movements including sprints, jumps, and shuffles, as well as skill-related actions with the ball [1]. The physical and physiological demands during competitions can be substantial, given the fast transitions between offense and defense [2,3]. Players take on specific positions based on their roles during gameplay. The most common position classification includes three (guards, forwards, and centers) or five roles (point guard, shooting guard, small forward, power forward, and center) [1,4]. With increasing specialization, some systems differentiate up to eight or thirteen positions [5,6]. Various methods have been utilized for playing position classification, such as game statistics analysis [7,8] or principal component analysis [6,9].

The increasing specialization of players into positions is closely tied to the gameplay dynamics and rules that characterize high-level basketball [4]. International governing body FIBA dictates regulations that shape spacing, timeouts, substitutions, and other structural elements that compel teams to strategize roles [10]. At the elite professional level, tailored tactics maximizing each position’s efficacy based on their court movement and actions are paramount. In this sense, players in different basketball positions present different characteristics: (a) guards are shorter, lighter and rely more on speed, agility and shooting skills; (b) forwards are versatile players who operate closer to the basket and are involved in more rebounding plays; and (c) centers are the tallest players, focusing on post play near the basket and defense around the paint [8,11]. These distinct roles result in variability in movement patterns and physiological responses [1,12]. Guards and forwards covered more distance, changes in direction, high-intensity sprints, and dribbles than centers [3,13]. The greater distances covered and accelerations performed by guards contribute to a higher external workload [3,14]. Otherwise, centers received more impacts, collisions, and contact with opponents [15,16]. The differences extend to internal workload indicators, such as heart rate responses, with higher values for guards than forwards and centers suggesting greater cardiovascular intensity [1,17]. Identifying position-specific demands allows the individualization of training programs to optimize performance during games [18].

Player tracking systems and microtechnology are often used to monitor external workload variables across playing positions, including the distance covered, accelerations/decelerations, impacts, and player load [19,20]. To classify the intensity of movements, commonly, five ranges have been utilized (e.g., covered distance: standing, walking, jogging, running, and sprinting) [1,3]. These intensity thresholds are generic for the device manufacturer (e.g., covered distance: standing >6 km/h, walking 6–12 km/h, jogging: 12–18 km/h, running 18–24 km/h, and sprinting >24 km/h) [17,21]. However, these predetermined thresholds may not accurately reflect the real demands of experienced basketball players due to the maximum speed registered in competition being between 18 and 22 km/h [3,22]. In this sense, it is necessary to improve the methodological aspects of using individualized thresholds in team sports [23,24].

To address the abovementioned issue, different methods have been proposed to calculate intensity thresholds in team sports based on maximum values, using Gaussian distributions, k-means clustering, or spectral clustering [25]. In basketball, k-means clustering and two-step clustering have been utilized to individualize intensity thresholds of distance covered, accelerations, decelerations, impacts, and player load in male and female basketball players [26,27]. The advantage of two-step clustering is the robust mathematical method that automatically provides a number of thresholds, while k-means clustering allows a set of five thresholds based on expert judgment and the previous literature [27,28]. However, these studies have established external load intensity ranges considering the whole team.

Since previous research has found differences in the external load performed based on the specific playing position of the players, as they have different roles during the game [1,3,12], an improved understanding of position-specific intensity thresholds can optimize training and game management. Therefore, the present study aims to describe and establish intensity thresholds based on mathematical models (k-means clustering and two-step clustering) of external load variables, considering the specific playing position of basketball players, as well as to compare the results obtained to determine which mathematical model is more useful in classifying external demands.

## 2. Materials and Methods

### 2.1. Study Design

The present study aims to establish position-specific intensity thresholds and determine differences in external workload demands during basketball games across playing positions. To test this, an observational quantitative study design was utilized, given the inability to actively manipulate or intervene in official basketball competitions [29]. The independent variable was playing position, with three levels: guards, forwards, and centers. The dependent variables were external workload metrics monitored via inertial devices during live games, including speed, accelerations, decelerations, impacts per minute, and player load per minute. These specific external workload variables were selected as key indicators of match physical demands based on the previous basketball literature [1,3]. The continuous tracking of these variables allows the quantification of external workload profiles by positions. Furthermore, established mathematical clustering techniques like two-step clustering and k-means clustering can categorize the external workload data into zones to derive intensity thresholds fitted to each playing position [25,26]. The findings can elucidate disparities between playing positions and establish more targeted, individualized intensity thresholds for optimal training and workload management in basketball.

### 2.2. Participants and Sample

We included 68 professional male basketball players from 6 different teams competing in the top-tier Spanish ACB league during the 2022–2023 preseason. Data were collected during six official preseason games involving ACB teams. It is important to mention that the Spanish basketball first division regulations prohibit inertial devices during official in-season competitions, so data had to be registered in two official preseason tournaments. To be included in the study, players had to meet the following criteria: (a) belong to the official roster of their respective first team in the ACB league; (b) play at least 5 min during the preseason tournaments analyzed; (c) have no musculoskeletal injuries in the 15 days prior to the games that would have limited their maximum performance; and (d) undergo a 10-day adaptation period with the inertial devices used for monitoring before data collection.

The study obtained approval from the University Bioethics Committee (233/2019) and adhered to the ethical guidelines outlined in the Declaration of Helsinki (2013). Participation was voluntary, with written informed consent obtained from all players. Coaching staff and team managers were informed about the study’s purpose, procedures, and potential risks and benefits prior to providing consent for their teams’ participation. Players were also briefed about the study details and provided consent before data collection. All participant data were anonymized, and confidentiality was maintained.

### 2.3. Variables and Equipment

To evaluate the external workload performed across playing positions during matches, five kinematic and neuromuscular variables were chosen that are commonly used by basketball teams according to previous research [19]: ***Velocity:*** the speed at which a player moves across the court, measured in kilometers per hour (km/h).***Acceleration:*** the rate at which a player increases his velocity, measured in meters per second squared (m/s^2^).***Deceleration:*** the rate at which a player decreases his velocity, measured in meters per second squared (m/s^2^).***Impacts/min:*** the number of times a player makes contact with another player or with the ground with a g-force higher than 1 g per minute. It is measured in counts per minute (n/min).***Player load/min:*** measurement derived from the accelerometer of the total body load in its 3 axes of movement (vertical, anteroposterior, and mediolateral), calculated as the square root of the sum of the accelerations divided by sampling frequency [30]. It is a sum of distance covered, accelerations and decelerations, and impacts and is measured in arbitrary units per minute (a.u./min).

To obtain the external workload variables, all basketball players were equipped with WIMU PRO^TM^ inertial devices (RealTrack Systems, Almería, Spain), which were located on the player’s back (between T2 and T4 at intra-scapular level) and fitted using an anatomical harness. To record kinematical variables (speed, accelerations, and decelerations), the inertial devices are equipped with ultra-wideband technology (UWB) for recording in indoor environments at a frequency of 33 Hz. The UWB system consisted of eight antennae that were placed around the court following the recommendations provided by Pino-Ortega et al. [31] for switching on and calibrating to guarantee the reliability and validity of the measurements. On the other hand, to record neuromuscular variables (impacts and player load), the inertial devices were equipped with different microsensors (four tri-axial accelerometers, three gyroscopes, and one magnetometer) that were set at 100 Hz [30].

### 2.4. Procedures

Firstly, the clubs were contacted to provide a clear understanding of the research objectives, along with the potential benefits and risks for the athletes involved. Upon obtaining consent from the clubs, further engagement was undertaken with the tournament organizers to seek authorization for the placement of UWB equipment on the playing fields. Once the proposal was approved, informed consent forms were signed by the coaches and players of the participating clubs.

Then, data collection was performed during two preparatory tournaments (six official games) of the ACB League, Spain’s top professional basketball league. To gather the necessary data, all players were equipped with WIMU PRO^TM^ inertial devices (RealTrack Systems, Almeria, Spain) using an anatomical harness. To minimize disruption to the players’ pre-match preparations, the equipment was fitted upon their arrival at the venue. The placement and calibration of the UWB system, as well as microsensors of inertial devices, took place 90 min before the start of each match, ensuring no interference with team preparations.

Following the conclusion of each match, data were extracted from the inertial devices using specialized software, SPRO^TM^ (v. 990, RealTrack Systems, Almeria, Spain). Subsequent analysis involved selecting relevant time sequences during which players were actively involved in the matches. The raw data were then exported to an Excel spreadsheet to create the database. Finally, data analysis was performed with statistical software, utilizing two methodologies: (a) unconditional analysis employing two-step clustering; and (b) widely adapted five-group k-means clustering algorithm. These approaches have been commonly employed in the existing literature to date, allowing for meaningful grouping and interpretation of the results [25,27,28].

### 2.5. Statistical Analysis

Raw data for the speed (km/h), positive and negative change in speed (accelerations/decelerations, m/s^2^), the impacts (count/min), and player load per minute (a.u./min) generated by all players during the matches were imported into the statistical package. These data generated a total sample of 3,345,703 cases for the speed variable, 22,943,386 cases for the acceleration variable, 26,451,233 cases for the deceleration variable, and 257 cases for the impacts and player load variables.

Then, two cluster analyses were performed with each of the five variables selected in this study: (a) two-step clustering to identify the load zones automatically; and (b) five-range k-means clustering. Previous research has employed these mathematical methods to classify intensity thresholds in basketball [26,27]. After identifying the centers of each cluster, a new variable identifying the membership of each case was generated to identify the lower and upper values and thus determine the working area of each variable. An ANOVA was performed to identify the existence of differences between each of the identified clusters. Data analysis was performed using the Statistical Package for the Social Sciences (SPSS, IBM, SPSS Statistics, v.25.0, Armonk, NY, USA). Statistical differences were considered if *p* < 0.05.

## 3. Results

### 3.1. Velocity by Playing Positions

Table 1 and Table 2 show the two-step clustering and k-means clustering analysis of velocity by playing positions during matches. Two-step clustering indicates two ranges in guards (<4.75 and >4.76 km/h) and three ranges in forwards (<2.74, 2.75 to 8.81, >8.82 km/h) and centers (<2.70, 2.71 to 8.69, >8.70 km/h). The highest distance in all positions was covered at low intensity (guards: 68.8%, forwards: 49.1%, and centers: 50.4%).

Otherwise, k-means clustering presents two zones over than the ranges provided by two-step clustering (running: 12.72 to 17.50 km/h; and sprinting: >17.51 km/h). Regarding k-means clustering, a higher percentage of standing and walking demands were performed by forwards and centers, while a higher percentage of jogging, running, and sprinting demands were performed by guards. ANOVA identified statistical differences in five-range k-means clustering in the total of cases (*F =* 10,027,814.10; *p* < 0.001) and per playing positions (guards: *F =* 2,493,424.24, *p* < 0.001; forwards: *F =* 4,578,381.41, *p* < 0.001; centers: *F =* 3,789,342.25, *p* < 0.001).

### 3.2. Changes in Speed by Playing Positions

The two-step clustering and k-means clustering analysis of speed changes (accelerations and decelerations) by playing positions during matches are shown in Table 3 and Table 4. Two-step clustering of accelerations indicates five ranges in guards (<0.31, 0.32 to 0.68, 0.69 to 1.23, 1.24 to 2.44, >2.45 m/s^2^), four ranges in forwards (<0.50, 0.51 to 1.10, 1.11 to 2.45, >2.46 m/s^2^), and three ranges in centers (<0.55, 0.56 to 1.34, >1.35 m/s^2^). Only guards and forwards presented a very high acceleration zone, with 1.3 and 1.1% of total accelerations. The two-step clustering of decelerations indicates three ranges in guards (>−0.48, −0.49 to −1.28, <−1.29 m/s^2^), and four ranges in forwards (>−0.35, −0.36 to −0.79, −0.80 to −1.64, <−1.65 m/s^2^) and centers (>−0.29, −0.30 to −0.67, −0.68 to −1.26, <−1.27 m/s^2^).

K-means clustering presents two zones over the ranges provided by two-step clustering in accelerations (high, 5.32–12.25 m/s^2^; very high, >12.26 m/s^2^) and decelerations (high, −2.99 to −6.27 m/s^2^; very high, <−6.28 m/s^2^), but these zones did not represent more than 1%. The majority of accelerations and decelerations are between the very-low and low categories (>96%). ANOVA identified statistical differences in five-range k-means clustering in the total of cases (*F =* 10,460,669.16; *p* < 0.001) and per playing position (guards: *F =* 3,066,120.22, *p* < 0.001; forwards: *F =* 5,612,608.43, *p* < 0.001; centers: *F =* 4,106,991.93, *p* < 0.001).

### 3.3. Impacts per Minute by Playing Position

Table 5 and Table 6 show the two-step clustering and k-means clustering analysis of impacts per minute by playing positions during matches. Two-step clustering indicates three ranges in guards (<57.65, 88.63 to 139.27, >143.11 n/min) and forwards (<125.52, 128.10 to 158.09, >161.83 n/min), and two ranges in centers (<142.00, >143.45 n/min).

K-means clustering presents a very low zone that was not detected by two-step clustering. In addition, a higher percentage of the moderate numbers of impacts per minute was performed by forwards and centers while a higher percentage of impacts by centers were performed in high and very-high zones. ANOVA identified statistical differences in five-range k-means clustering in the total cases (*F =* 593.048; *p* < 0.001) and per playing position (guards: *F =* 114.19, *p* < 0.001; forwards: *F =* 298.24, *p* < 0.001; centers: *F =* 246.51, *p* < 0.001).

### 3.4. Player Load per Minute by Playing Position

The two-step clustering and k-means clustering analysis of player load per minute by playing position during matches are shown in Table 7 and Table 8. The two-step clustering of accelerations indicates two ranges in all playing positions (guards: <1.10, >1.11 a.u./min; forwards: <1.25, >1.26 a.u./min; centers: <1.11, >1.12 a.u./min). K-means clustering presents three zones (very low, low, and high), more than the ranges provided by two-step clustering. The majority of player load per minute accumulated by centers and forwards is with low and moderate magnitude, while guards have high and very-high magnitude. ANOVA identified statistical differences in five-range k-means clustering in the total cases (*F =* 600.956; *p* < 0.001) and per playing position (guards: *F =* 94.23, *p* < 0.001; forwards: *F =* 204.46, *p* < 0.001; centers: *F =* 253.43, *p* < 0.001).

## 4. Discussion

This research aimed to find out the different kinematic and neuromuscular load zones according to specific positions in basketball players. The present study revealed clear differences in external workload demands during basketball games based on playing positions. Guards covered a greater total distance, especially at high intensities like jogging, running, and sprinting. This aligns with previous research showing guards performing more high-intensity running and changes in direction [3]. The need to frequently transition from offense to defense places extensive movement demands on guards. In contrast, centers and forwards exhibited lower speed profiles, concentrating movement close to the basket. Centers, in particular, operated predominantly in the key area, as evidenced by the highest impacts per minute. Their role revolves around screening, boxing out, and defending the paint [1,32]. These disparities in game demands likely stimulate position-specific physiological responses. For example, guards demonstrate higher heart rate intensity, reflecting greater cardiovascular demands from constantly moving up and down the court [17]. Identifying differences between positions is crucial for optimizing training and workload management, as this makes it possible to personalize training according to the specificity required for each variable.

The personalization of training is increasingly vital in team sports like basketball. While the training regimen has a collective objective requiring cooperation, each player must develop position-specific skills in order for the team to withstand diverse competitive challenges. Kozina et al. determined that training individualization is necessary, and systems should facilitate coaches’ work [33]. After a systematic review, Reina et al. [12] recommended personalized training based on the specific demands of each basketball position, supported by research on position-specific loads [11,34]. Therefore, identifying demands by position is key since roles require predominant technical/tactical actions that must be considered by coaches for the personalized preparation of players for games.

To establish protocols or reference values to individualize training, various statistical methods have been employed that allow the grouping of variables [11], verify whether statistical differences exist between specific positions based on load variables [3,12], or divide the load of different variables into correlative ranges to establish the optimal work zones [26]. One of the most-used analyses is utilized to determine reference values and to divide the variables into statistically different groups in their different modalities: hierarchical [35], k-means [26], and two steps [36]. The two-step classification method selects the number of groups based on the statistical differences that exist, without the researcher being able to intervene. This classification method sometimes has a low applicability, since the discrimination of the loads that coaches require does not occur [27]. The use of k-means analysis allows the pre-establishing of groups according to scientific evidence, allowing greater contextualization and applicability to established thresholds and thus improving the training process.

The results of the two-step classification method for the speed and acceleration variables show different groupings based on specific positions. The number of groups identified and the thresholds of each group are not directly applicable or comparable to the thresholds presented in the literature [27], but it is useful to know the descriptive behavior of the different specific positions of basketball players. The identification of two speed groups for the guards, and three groups for the forwards and pivots, shows that the guards have less variation in speed. During games, guards set the pace of the game and the construction of the play, trying to play fast or slow since they usually work under predetermined plays before the game [16]. On the other hand, forwards and pivots have greater oscillations in speed because they are more reactive groups that have to interact with the environment to make decisions to try to surprise the rival [10,37]. Regarding acceleration and deceleration, this classification is inverted. The guards have a greater number of zones because they have a greater variety of different movements by having more space on the field and participate more in the play [14], while pivots have less space and a narrower range of motions, executing more defined movements, focused essentially on enhancing their speed and agility for gestures such as jumping to shoot or rebound [37].

In the comparison between specific positions with the results of the k-means cluster, differences are found between specific positions. The guards have lower values in the lower speed zones and higher values in the faster speed zones. This is also seen with the percentage distribution of speed values; the distribution is lower as the pace increases, but it is always higher in the guards than in the pivots. Guards are players who run and accelerate less than pivots but maintain these high values for longer. These results align with those found by Petway et al. [3], who detected that guards performed high-speed values during displacements and also achieved them in acceleration. The high-intensity load is distributed at a very low percentage in high-performance settings across various specific positions. These observations align with Scanlan et al. [38], who emphasize the intermittent nature of movements made in basketball, with an average of 1750 changes in speed between the different intensity zones throughout a game. In basketball, not only it is important to know the external load’s diverse intensity zones but it is crucial to be aware of the speed distribution in each of the zones and the behaviors that lead athletes to different intensities in order to plan training and prepare players to withstand the demands of competition [18,26,39].

Homogeneity characterizes the neuromuscular load classifications for player load and impacts, with comparable groupings within variables across positions. Previous research found that loads were higher in competitions than training, with pivots incurring greater impact loads than forwards and guards [3,15]. This trend manifests in the present study’s results, revealing the two-step classification method’s impact distribution with higher values in pivots regarding intensity values. Moreover, the k-means cluster analysis evidences this difference, as pivots have elevated values in zones 4 and 5, indicating that they sustain more impacts than guards, with up to 45% of the impacts per minute relative to the total time-weighted impacts. Conversely, guards exhibit higher player load values than pivots both in men’s [2,3,27], and women’s basketball [12,15,26]. The findings again are related to the k-means cluster analysis, where guards display greater intensity across player load zones, accompanied by higher values in these zones. This leads to guards moving with higher absolute acceleration and more frequently than pivots [11,16]. These results represent the court functions performed by specific positions, influenced by tactics and players’ physical capacities [4]. While pivots’ loading arises from constant opponent contact and struggling to score near the basket, guards generate substantial physical loading by repeating highly explosive motions over time. Therefore, these aspects should prompt conditioning coaches to develop two targeted strategies for personalized work for training and recovery processes. 

Although the present study is the first approach to the classification of kinematic and neuromuscular demands in professional basketball by specific positions for the individualization of training loads and recovery processes, different limitations must be considered. First, the sample included only male professionals from a single league in preseason games. Evaluating female athletes and youth players would discern whether thresholds differ by sex and skill level. Similarly, analyzing data across full regular seasons with higher game intensity could refine and validate the reported thresholds. Second, the study derived thresholds based solely on external load metrics. A more comprehensive understanding should examine internal workload responses relative to the external classifications. Connecting physiological indicators like heart rate to the intensity zones would inform the true physical demands. Third, tactical and contextual factors like the game pace, match status, or attack–defense phase were not considered but may modulate workload outputs. Finally, the customized ranges provide objective classifications, but some overlap between positions still exists. Future approaches could apply more advanced machine learning techniques to improve the sensitivity and individual customization of thresholds. Overall, position-specific thresholds allow more effective workload monitoring to improve player health, performance, and training design.

## 5. Conclusions

The present study aimed to describe and establish intensity thresholds for common external workload variables in basketball based on playing positions. The results demonstrate clear differences in external load profiles between guards, forwards, and centers during competitions. Guards covered greater distance in high-intensity zones including jogging, running, and sprinting compared to forwards and centers. Guards also exhibited a higher number of accelerations and decelerations, though the magnitudes were predominantly low. Forwards and centers performed the majority of moderate speed activity. Centers experienced the highest frequency of impacts and collisions with other players. 

The findings highlight the disparities in game demands across playing positions arising from distinct roles during gameplay. Guards cover more of the court and execute quicker multidirectional movements, leading to greater acceleration/deceleration loads. The versatile roles of forwards position them closer to the basket, with more rebounding tasks. Centers operate predominantly around the key area with frequent box-outs, screens, and other contact actions.

Establishing intensity zones using mathematical techniques like k-means clustering and two-step clustering allows the individualization of thresholds fitted to each playing position. Though some overlap exists between positions, the tailored thresholds provide stronger practical utility compared to universal thresholds from device manufacturers. Teams can implement these findings to improve the monitoring of the external workload in basketball due to individualizing thresholds and can also enhance their analysis of competition performance and refine training programs tailored to game demands. On a practical level, coaches could apply these findings by:Using the thresholds as guidelines in training drills to expose players to competition intensity by position. During training, tasks will be designed to address specific demands per position, accounting for work thresholds and action quantities within each work range. For example, guards would perform more running and sprinting tasks, while large amounts of screening contact are implemented in center drills.Considering the thresholds when interpreting external loads from monitoring devices. A certain volume of impacts may signal high intensity for a guard but a normal range for centers during games due to different standards. Targeted thresholds facilitate more sensitive alert systems to prompt interventions around excessive loads and guide return-to-play protocols.Individualizing post-game recovery programming by prescribing active rest for positions accruing heavy accelerations/decelerations versus more passive modalities for those incurring extensive impacts. In this sense, tailored fitness programs will be created to enable players to recover from competition demands, compensating for produced imbalances.

## Figures and Tables

**Table 1 sensors-24-01146-t001:** Two-step clustering of velocity by playing positions.

Role	Speed (km/h)	Low/Walking	Moderate/Jogging	High/Sprinting
**Guard**	**Cluster Centers**	1.75	8.82	
	**Ranges**	<4.75	>4.76	
	**%**	68.8%	31.2%	
	**Distribution**	** 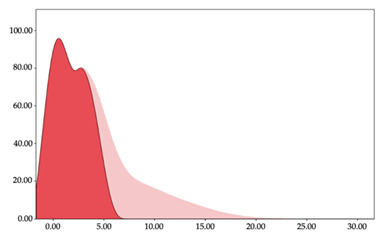 **	** 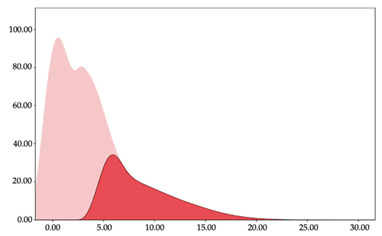 **	
**Forward**	**Centers**	0.84	4.92	12.87
	**Ranges**	<2.74	2.75 to 8.81	>8.82
	**%**	49.1%	37.1%	13.9%
	**Distribution**	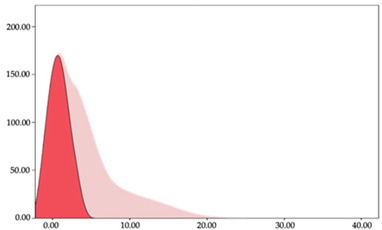	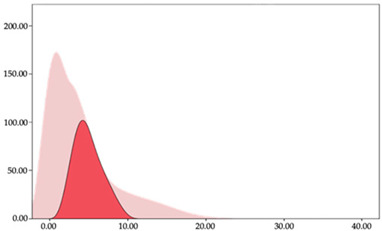	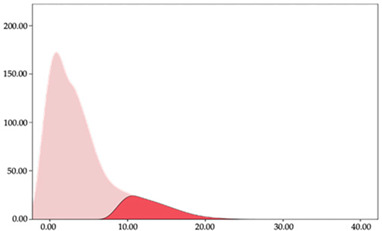
**Center**	**Centers**	0.80	4.86	12.74
	**Ranges**	<2.70	2.71 to 8.69	>8.70
	**%**	50.4%	36.8%	12.7%
	**Distribution**	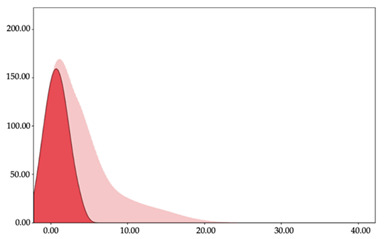	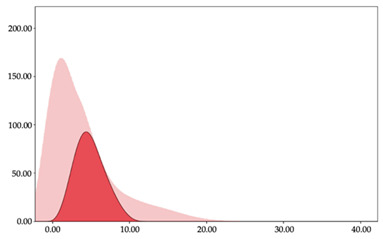	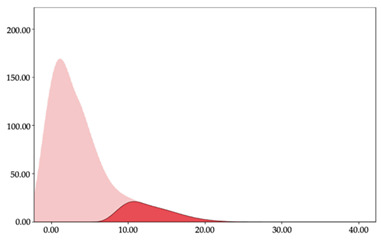

**Table 2 sensors-24-01146-t002:** Five-range k-means clustering of speed by playing positions.

Role	Speed (km/h)	Very Low/ Standing	Low/ Walking	Moderate/ Jogging	High/ Running	Very High/ Sprinting
**Guard**	**Centers**	0.70	4.12	8.26	12.61	17.33
	**Ranges**	<2.46	2.47 to 6.34	6.35 to 10.67	10.68 to 15.22	>15.23
	**%**	43.77%	35.35%	12.47%	6.39%	2.01%
**Forward**	**Centers**	0.81	4.40	9.09	13.81	18.57
	**Ranges**	<2.67	2.68 to 6.93	6.94 to 11.74	11.75 to 16.50	>16.51
	**%**	48.33%	32.56%	11.26%	6.02%	1.83%
**Centers**	**Centers**	0.87	4.67	9.88	15.17	20.06
	**Ranges**	<2.86	2.87 to 7.51	7.52 to 12.97	12.98 to 18.42	>18.43
	**%**	52.27%	31.98%	10.52%	4.54%	0.69%
**Total**	**Centers**	0.92	4.73	9.85	14.70	19.35
	**Ranges**	<2.95	2.96 to 7.58	7.59 to 12.71	12.72 to 17.50	>17.51
	**%**	51.7%	32.0%	10.60%	4.60%	1.00%

**Table 3 sensors-24-01146-t003:** Two-step clustering of changes in speed (accelerations and decelerations) by playing positions.

Accelerations						
Role	Acc (m/s^2^)	Very Low	Low	Moderate	High	Very High
**Guard**	**Centers**	0.12	0.49	0.88	1.63	3.57
	**Ranges**	<0.31	0.32 to 0.68	0.69 to 1.23	1.24 to 2.44	>2.45
	**%**	34.5%	23.1%	34.8%	6.4%	1.3%
	**Distribution**	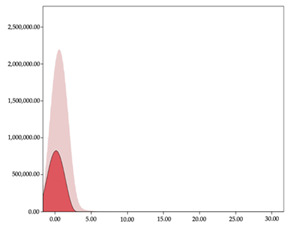	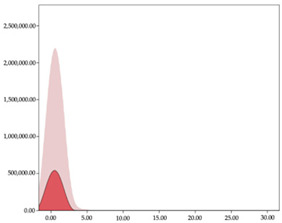	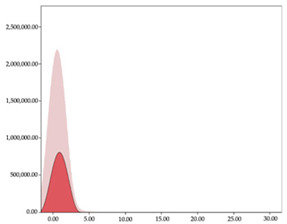	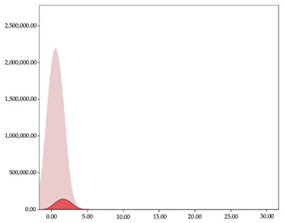	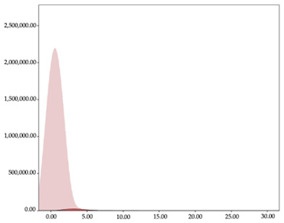
**Forward**	**Centers**		0.20	0.78	1.5	3.59
	**Ranges**		<0.50	0.51 to 1.10	1.11 to 2.45	>2.46
	**%**		43.5%	47.0%	8.4%	1.1%
	**Distribution**		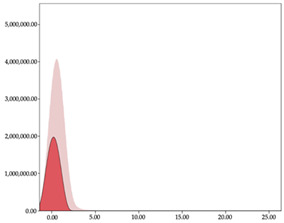	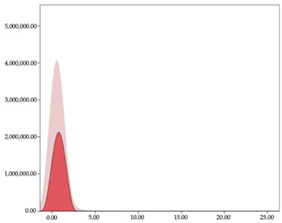	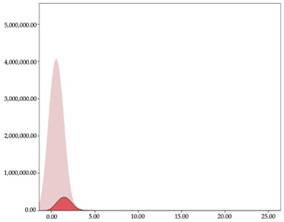	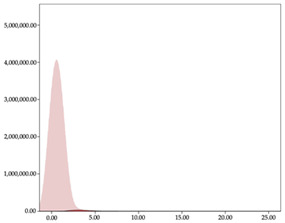
**Centers**	**Centers**		0.21	0.85	2.03	
	**Ranges**		<0.55	0.56 to 1.34	>1.35	
	**%**		47.3%	46.6%	6.1%	
	**Distribution**		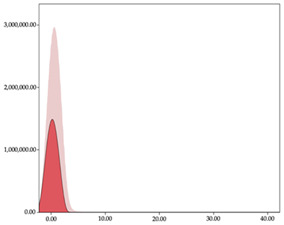	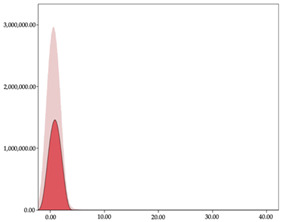	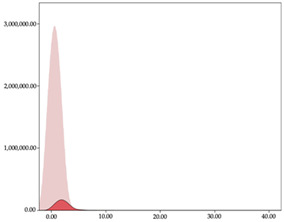	
**Decelerations**						
**Role**	**Dec (m/s^2^)**	**Very Low**	**Low**	**Moderate**	**High**	**Very High**
**Guard**	**Centers**	−0.22		−0.77	−1.86	
	**Ranges**	−0.48 to −0.00		−1.28 to −0.49	<−1.29	
	**%**	58.2%		37.8%	4.0%	
	**Distribution**	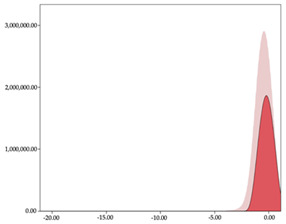		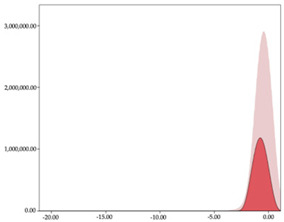	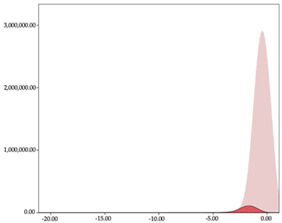	
**Forward**	**Centers**	−0.16	−0.55	−1.01	−2.20	
	**Ranges**	−0.35 to −0.00	−0.79 to −0.36	−1.64 to −0.80	<−1.65	
	**%**	42.7%	37.0%	18.2%	2.1%	
	**Distribution**	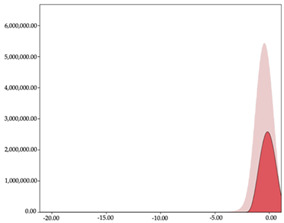	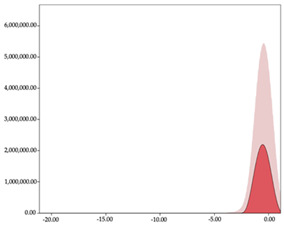	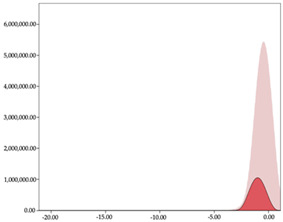	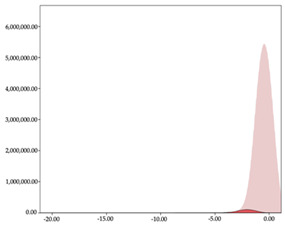	
**Centers**	**Centers**	−0.13	−0.46	−0.88	−1.72	
	**Ranges**	−0.29 to −0.00	−0.67 to −0.30	−1.26 to −0.68	<−1.27	
	**%**	37.4%	36.4%	22.5%	3.7%	
	**Distribution**	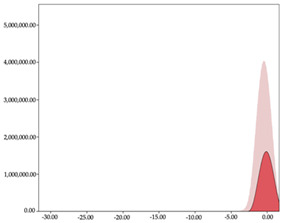	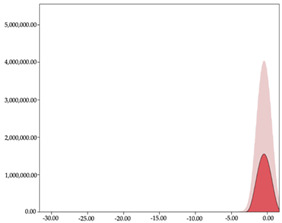	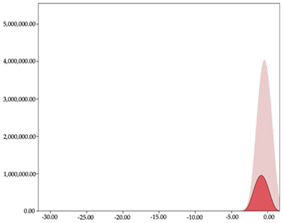	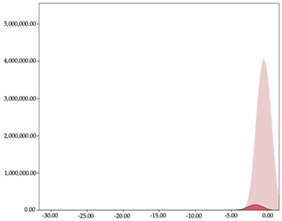	

**Table 4 sensors-24-01146-t004:** Five-range k-means clustering of speed by playing positions.

Accelerations						
Role	Acc (m/s^2^)	Very Low	Low	Moderate	High	Very High
**Guard**	**Centers**	0.45	1.36	3.49	6.80	13.84
	**Ranges**	<0.96	0.94 to 2.64	2.65 to 5.53	5.54 to 12.00	>12.01
	**%**	85.31%	13.66%	0.92%	0.10%	0.01%
**Forward**	**Centers**	0.39	1.09	2.86	5.53	9.19
	**Ranges**	<0.84	0.85 to 2.18	2.19 to 4.51	4.52 to 7.86	>7.87
	**%**	72.30%	26.20%	1.31%	0.18%	0.01%
**Centers**	**Centers**	0.44	1.24	3.40	6.84	25.33
	**Ranges**	<0.93	0.94 to 2.56	2.57 to 5.67	5.68 to 14.61	>14.61
	**%**	80.40%	18.63%	0.90%	0.06%	0.01%
**Total**	**Centers**	0.46	1.29	3.34	6.49	14.23
	**Ranges**	<0.95	0.96 to 2.53	2.54 to 5.31	5.32 to 12.25	>12.26
	**%**	83.49%	15.49%	0.93%	0.11%	0.06%
**Decelerations**						
**Role**	**Dec (m/s^2^)**	**Very Low**	**Low**	**Moderate**	**High**	**Very High**
**Guard**	**Centers**	−0.27	−0.86	−2.27	−4.39	−9.21
	**Ranges**	−0.59 to −0.00	−1.55 to −0.60	−3.51 to −1.59	−7.40 to −3.52	−18.99 to −7.41
	**%**	67.56%	30.01%	2.30%	0.12%	0.01%
**Forward**	**Centers**	−0.26	−0.86	−1.96	−4.27	−9.17
	**Ranges**	−0.59 to −0.00	−1.51 to −0.60	−3.44 to −1.52	−7.37 to −3.45	−19.45 to −7.38
	**%**	66.58%	30.76%	2.55%	0.10%	0.01%
**Centers**	**Centers**	−0.27	−0.87	−2.05	−5.62	−13.84
	**Ranges**	−0.59 to −0.00	−1.58 to −0.60	−4.32 to −1.59	−11.50 to −4.33	−26.03 to −11.51
	**%**	68.28%	30.04%	1.64%	0.03%	0.01%
**Total**	**Centers**	−0.26	−0.82	−1.78	−3.72	−7.91
	**Ranges**	−0.56 to 0.0	−1.37 to −0.57	−2.98 to −1.38	−6.27 to −2.99	−14.55 to −6.28
	**%**	64.34%	32.41%	3.07%	0.17%	0.01%

**Table 5 sensors-24-01146-t005:** Two-step clustering of impacts per minute by playing position.

Role	Imp (n/min)	Low	Moderate	High	Very High
**Guard**	**Centers**	42.58	119.60		160.63
	**Ranges**	<57.65	88.63 to 139.27		>143.11
	**%**	3.8%	54.7%		41.5%
	**Distribution**	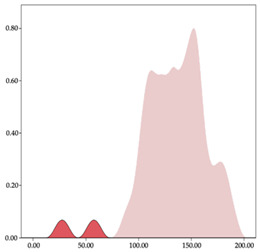	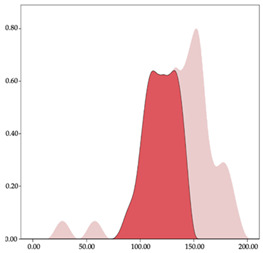		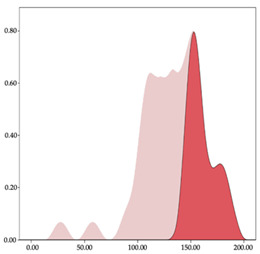
**Forwards**	**Centers**		111.78	142.22	174.85
	**Ranges**		<125.52	128.10 to 158.09	>161.83
	**%**		31.3%	53.5%	15.2%
	**Distribution**		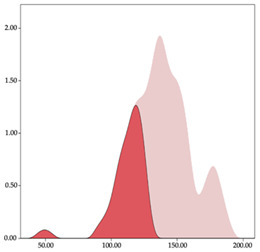	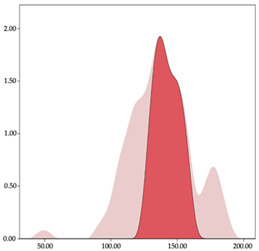	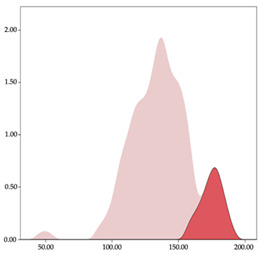
**Centers**	**Centers**		114.81		168.64
	**Ranges**		<142.00		>143.45
	**%**		55.2%		44.8%
	**Distribution**		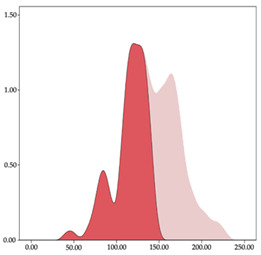		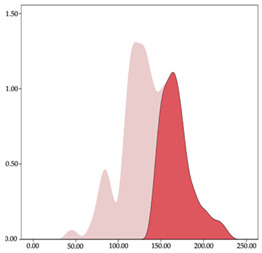

**Table 6 sensors-24-01146-t006:** Five-range k-means clustering of impacts per minute by playing position.

Role	Imp (n/min)	Very Low	Low	Moderate	High	Very High
**Guard**	**Centers**	27.50	57.65	88.63	139.27	188.10
	**Ranges**	<27.50	57.65 to 88.62	88.63 to 127.14	129.04 to 157.44	160.31 to 188.10
	**%**	1.89%	1.89%	12.47%	41.51%	16.98%
**Forward**	**Centers**	49.27	93.75	125.52	163.78	186
	**Ranges**	<49.27	93.75 to 119.23	120.56 to 139.67	140.63 to 163.78	171.75 to 186.00
	**%**	1.01%	21.21%	33.33%	32.32%	12.12%
**Centers**	**Centers**	45.33	78	123.09	188.1	223.47
	**Ranges**	<45.33	68.77 to 104.44	108.33 to 142.00	143.45 to 177.67	181.41 to 223.47
	**%**	0.95%	11.43%	42.86%	34.29%	10.48%
**Total**	**Centers**	57.45	107.4	132.01	155.1	182.69
	**Ranges**	<78.00	83.22 to 119.60	120.53 to 143.60	143.81 to 169.14	169.43 to 223.47
	**%**	2.72%	23.74%	33.07%	26.46%	14.01%

**Table 7 sensors-24-01146-t007:** Two-step clustering of player load per minute by playing position.

Role	PL (a.u./min)	Moderate	High
**Guards**	**Centers**	0.89	1.37
	**Ranges**	0.19 to 1.10	1.11 to 1.88
	**%**	35.8%	64.2%
	**Distribution**	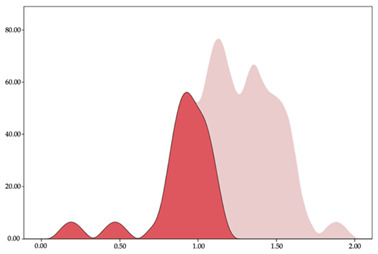	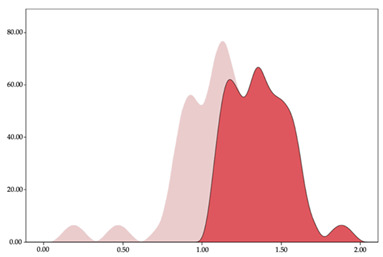
**Forwards**	**Centers**	1.06	1.44
	**Ranges**	0.38 to 1.25	1.26 to 1.93
	**%**	64.6%	35.4%
	**Distribution**	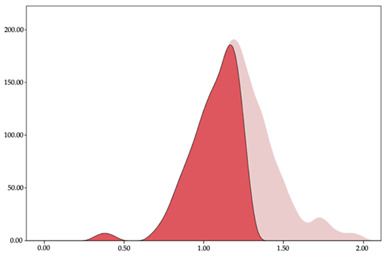	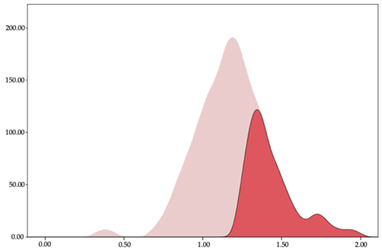
**Centers**	**Centers**	0.92	1.33
	**Ranges**	0.27 to 1.11	1.12 to 1.83
	**%**	56.2%	43.8%
	**Distribution**	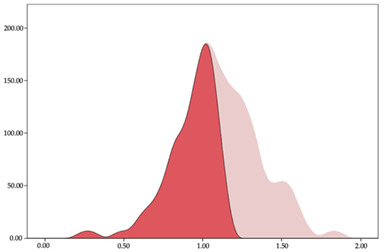	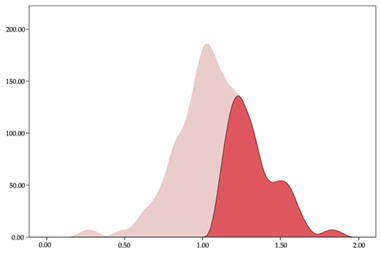

**Table 8 sensors-24-01146-t008:** Five-range k-means clustering of player load per minute by playing positions.

Role	Imp (n/min)	Very Low	Low	Moderate	High	Very High
**Guard**	**Centers**	0.19	0.47	0.93	1.21	1.52
	**Ranges**	0.19 to 0.47	0.48 to 1.06	1.07 to 1.33	1.34 to 1.64	1.65 to 1.88
	**%**	1.89%	1.89%	28.30%	35.85%	32.08%
**Forward**	**Centers**	0.38	0.96	1.21	1.43	1.77
	**Ranges**	0.38 to 0.38	0.39 to 1.03	1.04 to 1.25	1.26 to 1.48	1.49 to 1.93
	**%**	1.01%	31.31%	43.43%	19.19%	5.05%
**Centers**	**Centers**	0.56	0.93	1.20	1.49	1.83
	**Ranges**	0.27 to 0.50	0.51 to 0.91	0.92 to 1.14	1.15 to 1.40	1.41 to 1.83
	**%**	5.71%	40.95%	39.05%	13.33%	0.95%
**Total**	**Centers**	0.36	0.89	1.14	1.40	1.72
	**Ranges**	0.19 to 0.50	0.51 to 1.01	1.02 to 1.27	1.28 to 1.56	1.57 to 1.93
	**%**	1.95%	27.24%	39.30%	26.85%	4.67%

## Data Availability

The data presented in this study are available on request from the corresponding author. The data are not publicly available due to the Organic Law 3/2018, of 5 December, on the Protection of Personal Data and Guarantee of Digital Rights of the Government of Spain, which requires that this information must be in custody.

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
