# Peer review of "Intensity Thresholds for External Workload Demands in Basketball: Is Individualization Based on Playing Positions Necessary?"

_sensors, 2024, doi:10.3390/s24041146_

Round 1

Reviewer 1 Report

Comments and Suggestions for Authors

The article is an experimental study with elite basketball players, it has a direct impact thanks to the selected sample. The article has a proper storyline and links goals to results correctly.

However, as a reviewer, it is propose a series of elements that are necessary for its improvement as a scientific article.

The first idea is general and has to do with the way of playing basketball and the scope of the article's research. It should be more associated in the introduction of the article that the dynamics of the game, rules and rules of FIBA basketball make the profile of the player must be specialized, not because he runs more or has a greater number of impacts in the game, but because in elite basketball, in men it is played with game dynamics, Regulations and standards. An example is to indicate that a center in basketball is normal that he does not travel so much distance in elite basketball since his function is to perform movement and actions because it is shown that this way his effect on the game and result is more effective.

The second idea is based on the evidence presented in the information of the article that is descriptive at a general level of the teams but that perhaps should better categorize the results of the research since it would be very interesting to know if there are differences between a team that wins and another that does not win, apart from knowing how a basketball player moves and what effort he makes, it would be interesting to know if there are differences between the players who win a match and the players who lose.

Specifically, it is recommending the review of:

In line 257 It not showing that the reference [12] as applicable to the context of the paragraph

In line 265 the reference [33] does not argue the cardiovascular characteristics of point guards in basketball

In lines 297 and 302 It not showing that the reference [38] as applicable to the context of the paragraph

In line 339 this paragraph in the reviewer's opinion should be in conclusions or contributions, rather than in a discussion.

The article reaches the standard of being publishable on sensors, if the reviewer's proposals are addressed and discussed.

Regards

Author Response

Dear Reviewer,

Firstly, thank you for your words and time spent reviewing this article. We have carefully considered all reviewers' considerations of the paper (sensors-2815711). Please find enclosed our detailed answers to reviewers' queries in attached document. This document will include your appreciations and the modifications made by the authors considering your comments.

We hope that the changes will be in accordance with your expectations. Your contributions have improved the quality of the article considerably. To facilitate your work, all corrections to the article are shown in red. The authors declare that the manuscript is original and has not been considered for publication elsewhere. Additionally, the authors had approved the paper for release and agree with its content.

Kindly regards

Reviewer 2 Report

Comments and Suggestions for Authors

I want to thank you for the opportunity to review this manuscript. In general, this is in advance, really interesting research about external workload demands in basketball. After a careful analysis, I believe that this paper, in my opinion, is ready for publication. Therefore, I hope my few recommendations will help you improve the manuscript.

Abstract:

The abstract is very clear and complete the only question is if the preseason official games intensity is similar to season games, this can introduce a bias in your research.

Introduction:

Is well structured and presents enough and recent background to introduce the research aims.

Methods:

Maybe, you can include this text in 2.2. Participants and sample “It is important to mention that the Spanish basketball first division regulations prohibit inertial devices during official in-season competitions, so data had to be registered in two official preseason tournaments” instead of in point 2.4. I could clarify and justify better the sample selection.

Raw data and the statistical procedure are clear and adequate to the aims of the paper

Results:

These very interesting results and perfectly presented.

Discussion:

Is well structured, very detailed, and accurate to the aims of the paper.

Conclusion:

In the abstract, you wrote “These data are highly applicable to the design of training tasks at the highest competitive level”, but in conclusions chapter, this phrase does not have the expected impact. Maybe you can include some hints/tips for trainers in order to help trainers with no experience in scientific data interpretation. This question is just a suggestion.

I hope my suggestions can improve your paper.

Kindest Regards

Author Response

Dear Reviewer,

Thank you for your words and time spent reviewing this article. We have carefully considered all reviewers' considerations of the paper (sensors-2815711). Please find enclosed our detailed answers to reviewers' queries. The attached document will include your appreciations and the modifications made by the authors considering your comments.

We hope that the changes will be in accordance with your expectations. Your contributions have improved the quality of the article considerably. To facilitate your work, all corrections to the article are shown in red. The authors declare that the manuscript is original and has not been considered for publication elsewhere. Additionally, the authors had approved the paper for release and agree with its content.

Kindly regards
